# Paranoia, self-deception and overconfidence

**Rosa A. Rossi-Goldthorpe**[1,2], **Yuan Chang Leong**[3], **Pantelis Leptourgos**[1], **Philip R. Corlett**[1,4]*

**1** Department of Psychiatry, Yale University, New Haven, Connecticut, United States of America,
**2** Interdepartmental Neuroscience Program, Yale University, New Haven, Connecticut, United States of America, **3** Department of Psychology, University of Chicago, Chicago, Illinois, United States of America, **4** Wu Tsai Institute, Yale University, New Haven, Connecticut, United States of America

* philip.corlett@yale.edu

**Data Availability Statement:** Data Availability: The data is available at https://github.com/rosarossig/self-deception.git. Code Availability: Code for the specific 2-stream HGF is freely available at https://osf.io/8kfph/.

## Abstract

Self-deception, paranoia, and overconfidence involve misbeliefs about the self, others, and world. They are often considered mistaken. Here we explore whether they might be adaptive, and further, whether they might be explicable in Bayesian terms. We administered a difficult perceptual judgment task with and without social influence (suggestions from a cooperating or competing partner). Crucially, the social influence was uninformative. We found that participants heeded the suggestions most under the most uncertain conditions and that they did so with high confidence, particularly if they were more paranoid. Model fitting to participant behavior revealed that their prior beliefs changed depending on whether the partner was a collaborator or competitor, however, those beliefs did not differ as a function of paranoia. Instead, paranoia, self-deception, and overconfidence were associated with participants' perceived instability of their own performance. These data are consistent with the idea that self-deception, paranoia, and overconfidence flourish under uncertainty, and have their roots in low self-esteem, rather than excessive social concern. The model suggests that spurious beliefs can have value–self-deception is irrational yet can facilitate optimal behavior. This occurs even at the expense of monetary rewards, perhaps explaining why self-deception and paranoia contribute to costly decisions which can spark financial crashes and devastating wars.

## Author summary

Paranoia is the belief that others intend to harm you. Some people think that paranoia evolved to serve a collational function and should thus be related to the mechanisms of group membership and reputation management. Others have argued that its roots are much more basic, being based instead in how the individual models and anticipates their world–even non-social things. To adjudicate we gave participants a difficult perceptual decision-making task, during which they received advice on what to decide from a partner, who was either a collaborator (in their group) or a competitor (outside of their group). Using computational modeling of participant choices which allowed us to estimate the role of social and non-social processes in the decision, we found that the manipulation worked: people placed a stronger prior weight on the advice from a collaborator

**Funding:** PRC was supported by the National Institute of Mental Health (R01MH12887). RR-G was supported by the Interdepartmental Neuroscience Program. This work was supported by the Yale University Department of Psychiatry (PRC), the Kavli Institute for Neuroscience Pilot Award (PRC), the Connecticut Mental Health Center (CMHC) and Connecticut State Department of Mental Health and Addiction Services (DMHAS) (PRC). The funders had no role in study design, data collection and analysis, decision to publish or preparation of the manuscript.

**Competing interests:** The authors have declared that no competing interests exist.

compared to a competitor. However, paranoia did not interact with this effect. Instead, paranoia was associated with participants' beliefs about their own performance. When those beliefs were poor, paranoid participants relied heavily on the advice, even when it contradicted the evidence. Thus, we find a mechanistic link between paranoia, self-deception and over confidence.

## Introduction

People lie to others, but they also lie to themselves. We might deceive others more convincingly by better deceiving ourselves [1]. Self-deception may also protect self-esteem [2]. We deceive ourselves into believing that we are kinder, fairer, and more proficient than average [1]. The accompanying overconfidence can be adaptive both intra- and interpersonally–increasing performance [2] and persuasiveness [3]. However, too much self-deception can culminate in deleterious consequences [4,5], and ultimately, delusional beliefs [6].

Paranoia–the belief that others have malicious intentions towards us–shares many of the hallmarks of self-deception [7]. It may protect self-esteem [8,9], and, by polarizing the social world it may solidify group identity [7], via direct inflation of self-image, or indirectly, through overconfidence. Confident people are more convincing [10,11], and, in so being, they further reinforce their own misbeliefs [12,13]. Paranoid beliefs are of course social, in that they are about powerful and nefarious others. The coalitional cognition account of paranoia posits that it arises from the excessive operation of an evolved mechanism of coalitional threat detection, which manages reputations and interactions with groups of others [7]. It has some support [14]. However, it is not clear that apparently complex social behaviors are necessarily underwritten by mechanisms dedicated to social cognition [15,16]. Paranoia was found to be unrelated to betrayal aversion–when one has a higher aversion to risky situations where outcomes are contingent upon social factors compared to non-social factors, which does not support the coalitional cognition model [17]. Instead, paranoia may arise from domain-general mechanisms of uncertainty weighted belief updating [18,19]. Here we attempted an explicit separation of social and non-social influences to belief updating and paranoia in order to shed light on whether paranoia arises from socially-specific processes or domain-general cognitive mechanisms. More broadly, given the potential social and non-social loci of self-deception, and the possible relationships between delusions and self-deception, we aimed to triangulate the relationships between paranoia, self-deception, and overconfidence, using a perceptual decision-making task, self-ratings of paranoia, and computational modeling of behavior. In so doing, we hoped to adjudicate between competing accounts. For example, we could relate paranoia, self-deception, and over confidence to the social processes in our task and model, and that would favor social accounts of these phenomena.

Self-deception flourishes under uncertainty [20], and in laboratory tasks, paranoid individuals expect more volatility but also fail to learn appropriately from volatility [18]. It is as yet unclear whether paranoia and self-deception share underlying psychological mechanisms, and whether they are similarly sensitive to uncertainty or social affiliative processes. A shared mechanism might suggest that paranoia could amplify self-deceptive behaviors, thus bolstering misbeliefs and causing more distress.

To investigate the relationships between paranoia and self-deception, we adapted a perceptual decision-making task with varying levels of stimulus ambiguity. The task has two sources of information, one social and one non-social, that can allow us to dissect differential contributions to the decision-making and explore interactions with paranoia. Using computational

modeling that explicitly quantifies these contributions of social and non-social information to decisions, we sought to delineate whether and how self-deception and over-confidence are related to paranoia. We hypothesized that paranoia would be associated with enhanced self-deception, as well as higher confidence reported overall due to the shared characteristics and relationship with delusional beliefs. In prior work we showed non-social mechanisms contributed to paranoia, whilst others have posited a specifically social, coalitional mechanism. We sought to adjudicate by examining the impact of group identity on perceptual decision making. If group identity interacts with paranoia status then we would favor coalitional accounts. If instead non-social mechanisms prevail then we would favor a domain-general explanation of paranoia.

## Methods

### Ethics statement

All experiments were approved by the Yale University Human Investigation Committee. Written informed consent was provided by all participants.

### Behavioral task

Participants classified merged images of faces and scenes, as either containing more face or scene, and they expressed their confidence in their choice [21]. These "chimeric" images ranged from 100% face and 0% scene to 100% scene and 0% face over 80 trials (*C1 Phase–No Partner*). After each classification they rated their confidence about their decision on a 1–7 scale. Participants were required to answer each trial before proceeding to the next. After the 80 trials, they were informed they were either working with a partner who was either a collaborator (N = 329), or a competitor (N = 334), who would be placing bets on whether the next image would be mostly face or mostly house (Fig 1A) [21]. In the cooperation condition, the participant would receive a monetary bonus if their partner's bet was correct, in addition to the earnings from correctly classifying the image (10 cents if both them and the partner were correct). In the competition condition, the participants would lose money if their partner's bet was correct (if their classification was correct, 4 cents; if their classification was incorrect, 7 cents). The payoff matrix for each condition is given in Fig 1B. Participants are not told that the partner or opponent has been given any more information than them, and importantly, the bet is made before the image for the trial is shown. Crucially, the reward maximizing strategy is to classify the images correctly. Participants were informed that they would be compensated based upon how many images they classified correctly, and were told this on both phases of the experiment. The participants saw their partner's bet before seeing the image, before providing their classification and confidence again (C2 *Phase*). They classified the same images they saw in *C1*. In experiment 1, the bets in the C2 phase were correct exactly 50% of the time. Note that in the C1 phase participants only classified the image and there were no bets–the bets were added in the C2 phase.

### Questionnaires

Participants reported demographic information (age, gender, income, educational level, ethnicity, and race) as well as mental health questions (diagnosis, medication use), and completed the Revised Green et al. Paranoia Thoughts Scale (R-GPTS) [22], Beck's Anxiety Inventory (BAI) [23], Beck's Depression Inventory (BDI) [24]. We included free response questions to detect bot respondents. Participants who scored 11 or higher on the R-GPTS persecution scale were classified as high paranoia as this is the recommended clinical cutoff [22]. Participants

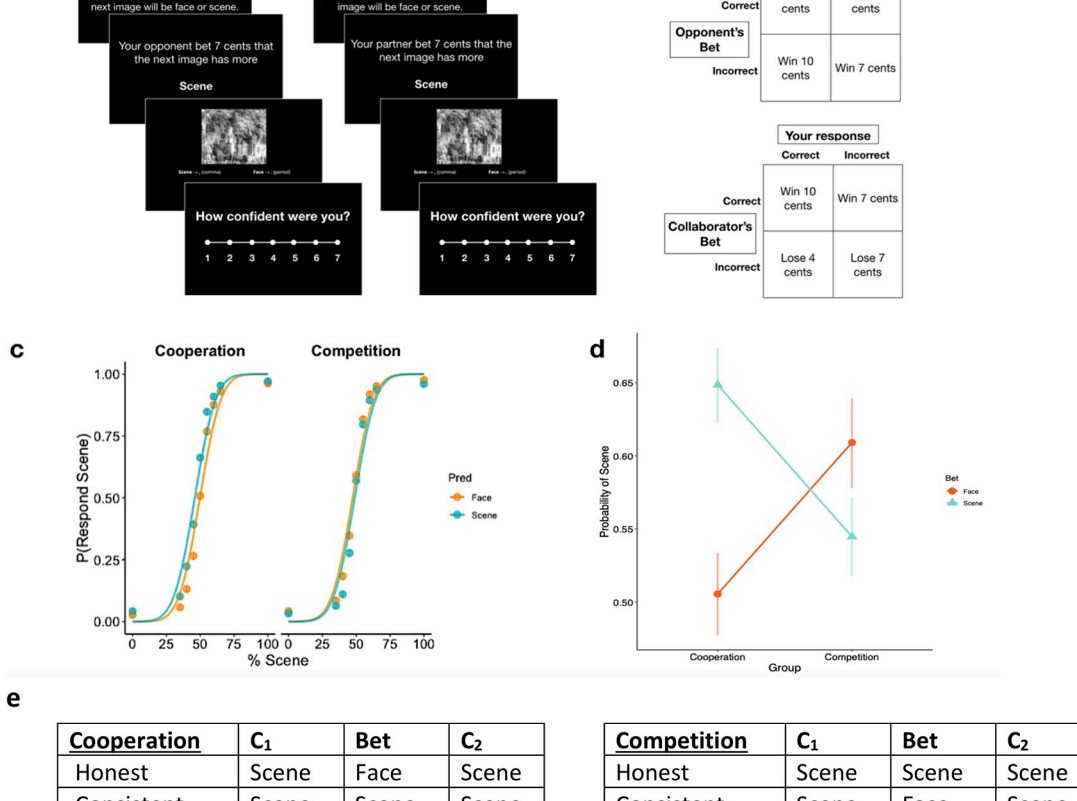

**Fig 1. Task structure for C2 phase and interaction effect. A**, sequence of task for the 2 conditions. **B**, payoff matrices for both conditions. Participants should ideally classify the image objectively (as they did in the initial classification phase) without using the bet to inform their decision. **C**, psychometric functions showing the percentage of scene in the image versus probability of responding scene averaged over all participants. **D**, participant's choices displayed a motivational bias. The bet x group interaction shows that participants in the cooperation group tended to align with the bet (higher probability of answering scene when the bet was scene), while the competition group tended to disagree with the bet (higher probability of responding with scene when the bet was face). **E**, response patterns for the two experimental conditions. Self-deception is defined differently based upon experimental group.

who scored above 16 on Beck's Anxiety Inventory were classified as high anxiety based on the recommended cutoff for clinically significant anxiety [25].

**Participants** (N = 719) were recruited for experiment 1 online via CloudResearch. Participants who declined more than 30% of the survey responses were automatically rejected. Non-sensical free responses were rejected (N = 48). For experiment 1, our total sample (N = 663) of complete submissions included 334 participants for the competition condition and 329 participants for the cooperation condition. For experiment 2, we applied the same criteria. Participants (N = 327) were recruited through CloudResearch, this time using the new Data Quality feature. Only 3 submissions were excluded.

## Behavioral analysis

Motivational bias was assessed with a general linear mixed effects model (GLME) using the lme4 package in R. GLMEs were also fit to choice data using only scene percentage as a

variable in order to confirm that classifications were related to the objective scene percentage (rather than random responding).

If a classification changed between sessions (*C1* and *C2*) to either agree with the bet (cooperation condition) or to disagree with the bet (competition condition) the response was self-deceptive [26]. Response patterns determining a self-deceptive trial are also shown in Fig 1E. The raw self-deception score for a participant was computed as the sum of the number of self-deceptive responses divided by the total number of responses. To explore whether participants were merely guessing when they changed their minds to conform to or defect from the bets, we multiplied their number of deception trials by their normalized confidence on those trials:

$$\text{CWSD} = \text{raw self}-\text{deception score} * \left( \frac{\text{mean confidence on self}-\text{deceptive trials}}{\text{mean confidence on C1 trials}} \right) \quad (1)$$

We will refer to this metric combining the amount of self-deception with the confidence increase while self-deceiving as confidence-weighted self-deception (CWSD).

## Computational modeling

We adapted open-access code for a Hierarchical Gaussian Filter (HGF) with 2 streams of processing in MATLAB 2018a (MathWorks R, Natick, MA). The HGF includes a generative model of the agent's inferences (perceptual model), and a response model incorporating their action choices. Our perceptual model had two layers of beliefs, split into separate social and non-social arms, and the response model was a softmax for binary choices [27] [28].

The first level of the generative model ($x_{1,s}$ and $x_{1,ns}$) represents the beliefs about the accuracy of the bet (1 = correct, 0 = incorrect) and the image category (1 = scene, 0 = face), respectively. The second level describes the perception of the tendency of the first level: the tendency for the bet to be correct ($x_{2,s}$) and the tendency of the image category ($x_{2,ns}$). The 2nd level has a Markov-like dependence where the estimate of $x_{2,s}$ and $x_{2,ns}$ are updated from their respective values on the previous time step according to a Gaussian random walk with variance ω:

$$x_{2,s}(t) \sim \mathcal{N}(x_{2,s}(t-1), \omega_s) \quad (2)$$

$$x_{2,ns}(t) \sim \mathcal{N}(x_{2,ns}(t-1), \omega_{ns}) \quad (3)$$

The first level beliefs are computed directly from 2nd level at time *t*, through a logistic sigmoid:

$$\hat{\mu}_{1,s}(t) = \frac{1}{1 + e^{-\eta\omega_s\mu_{2,s}(t-1)}} \quad (4)$$

$$\hat{\mu}_{1,ns}(t) = \frac{1}{1 + e^{-(\mu_{2,ns}(t-1)+\text{recency bias})}} \quad (5)$$

The specific formulations of Eqs 4 and 5 were deduced from model comparison. Since current classification might be influenced by the previous images, we incorporated a recency bias that weighted the non-social prediction towards the previous image, depending upon its ambiguity (Fig 2C). The recency bias is based upon the amount of ambiguity in the previous image. As a result, the recency bias towards a particular classification will be maximized when ambiguity is minimized. When the previous image is 50% face and 50% scene, the recency bias is zero. We map this recency bias to a linear function of the scene percentage of the image where

the maximum value is 1 and the minimum is -1:

$$\text{recency bias} = \frac{1}{50}(\text{scene percentage of image} - 50) \tag{6}$$

On the social side, we explored how adding a bias term on the logistic sigmoid connecting $x_{2,s}$ and $x_{1,s}$ might help explain motivated perception. We incorporated an additive term on the exponent (shifting the inflection point of the psychometric curve), a multiplicative term on the exponent (shifting the steepness of the psychometric curve) as well as a combination of those terms. The multiplicative term provided better fit, but we determined this term had a high correlation with $\omega_s$ (Pearson's r = 0.998, p <2.2–16). As a result, we replaced this multiplicative term with $\omega_s$ (perceptual model P3), and the best-fitting model had a bias term that was a linear scaling, $\eta\omega_s$ as a multiplicative term on the exponent (Fig 2B).

Although mapping the 2$^{nd}$ to first level was different between the two streams, the computations by which the beliefs evolved on the 2$^{nd}$ level were the same for the 2 processing streams.

The belief at the second level ($\mu_2$), is updated by the precision-weighted prediction error from the first level:

$$\Delta\mu_2(t) \propto \frac{\hat{\pi}_1(t)}{\pi_2(t)}\delta_1(t) \tag{7}$$

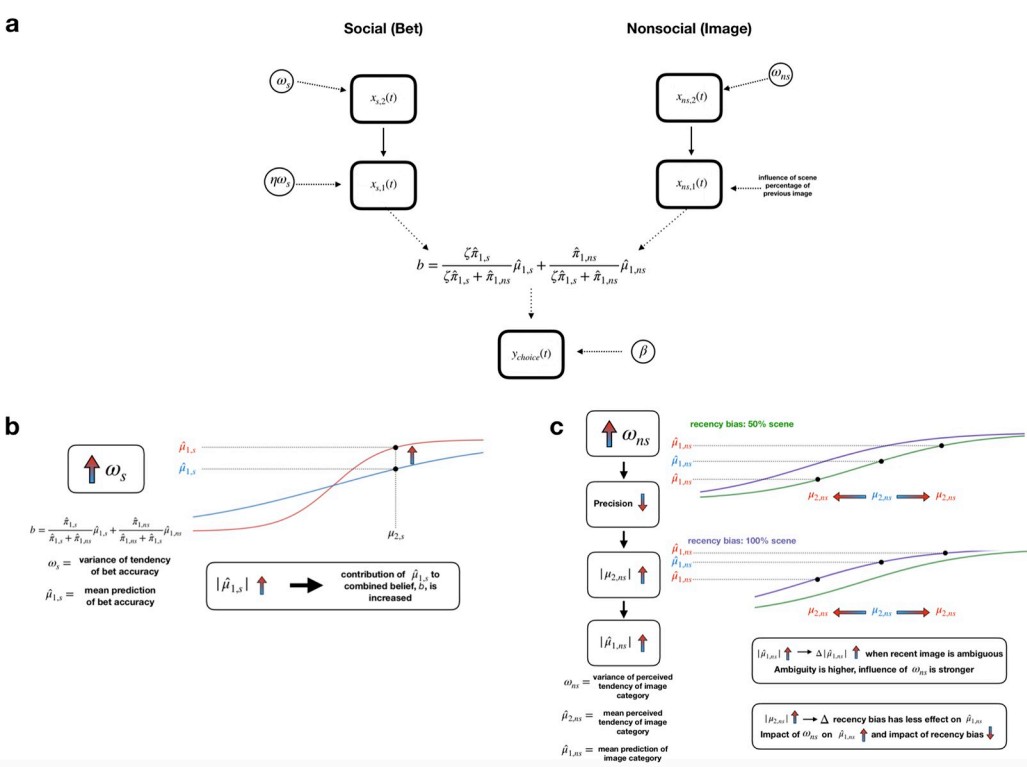

**Fig 2. A**, the 2-level HGF with parallel processing streams for social and non-social stimuli. The choice data is fed into the model, which is inverted to obtain parameter estimations for an individual. The perceptual model includes the both the social and non-social information, which is then used to compute the combined belief, $b$. This combined belief is the input to the response model. More details are given in the methods. **B**, increasing $\omega_s$ causes the prediction about the accuracy of the bet ($\hat{\mu}_{1,s}$) to become closer to an extreme (1 or 0). This tilts the combined belief towards this prediction. **C**, increasing $\omega_{ns}$ causes the recency bias to have less of an effect on the prediction about the image categorization ($\hat{\mu}_{1,ns}$) while perceived tendency (second level belief) dominates the prediction. The effect is stronger when the recent image is very ambiguous.

where $\delta_1$ is the prediction error at the first level and $\pi_2(t)$ is the precision of the posterior second level belief.

$$\delta_1(t) = \mu_1(t) - \hat{\mu}_1(t) \tag{8}$$

The first level predicted belief ($\hat{\mu}_1$) is determined by the logistic sigmoid above (Eqs 4 and 5), and the prediction error generated incorporates the model inputs (bet accuracy and scene percentage) for the respective processing streams for the current trial.

In order to combine the two information streams, the belief, $b(t)$, was computed as a linear combination of the predictions of the first level beliefs, weighted by their precisions.

$$b(t) = \frac{\hat{\pi}_{1,s}(t)\hat{\mu}_{1,s}(t) + \hat{\pi}_{1,ns}(t)\hat{\mu}_{1,ns}(t)}{\hat{\pi}_{1,s}(t) + \hat{\pi}_{1,ns}(t)} \tag{9}$$

This combined belief was then fed into a softmax function to compute the probability of agreeing with the bet:

$$P(y(t) = 1) = \frac{b^\beta}{b^\beta + (1-b)^\beta} \tag{10}$$

We also examined the effect of adding term to weight the two streams in the response model as in Diaconescu et al. (2014) [29] [27], which ultimately did not fit our behavioral data (S6 Table). Initial values for all parameters are in S7 Table.

## Statistics

Statistical analyses were performed in RStudio, Version 1.2.5033. Model parameters and self-deception scores were analyzed using ANOVAs, with Bonferroni correction for multiple-comparison (as needed). We performed ANCOVAs for model parameters using three sets of covariates: (1) demographics (age, gender, ethnicity, and race); (2) mental health factors (medication usage, diagnostic category); (3) and metrics and correlates of global cognitive function (educational attainment, income).

## Results

### Behavioral data

As the percentage of scene in the chimera increased, the probability of responding scene followed an s-shaped psychometric curve, indicating that in general, participants were able to categorize the chimeras accurately (Fig 1C). However, there was a motivational bias: the bets influenced the participants' choices differently based on experimental condition (cooperation vs. competition, a significant *bet x group* interaction, z = 8.802, p<2e-16, b = 0.131875, 95% CI: [0.1025, 0.16124]). Participants in the cooperation condition were more likely to agree with the bet while participants in the competition condition were more likely to disagree with the bet (Fig 1D), indicating that participants were motivated to respond based on their relationship with the partner.

### Paranoia and self-deception

We defined a self-deceptive response as a change in response between sessions C1 and C2 to either agree with the bet of the collaborator (cooperation condition), or to disagree with the bet of the opponent (competition condition). For each participant, the raw self-deception score was computed as the sum of the number of self-deceptive response. Using the response pattern of self-deception (Fig 1E) as well as our confidence-weighted self-deception metric

(Eq 1), we investigated the relationship between self-deception and paranoia. Analysis of variance revealed a main effect of paranoia (high or low) on self-deception scores, a main effect of group (competition or collaboration) but no paranoia by group interaction for self-deception. High paranoia participants made more self-deceptive choices (Self-deception score; $F(1, 659) = 13.65$, $p_{bonf} = 0.0007155$, $\eta_p^2 = 0.02045$), and were more confident on those trials (Mean confidence on SD trials; $F(1,620) = 81.691$, $p_{bonf} < 2e\text{-}16$, $\eta_p^2 = 0.116$). The difference between groups remained significant when we examined confidence-normalized self-deception score (CWSD equation; see Methods; $F(1, 620) = 58.0612$, $p_{bonf} = 2.8659e\text{-}13$, $\eta_p^2 = 0.0859$; Fig 3B and 3C). We also found that the cooperation group had increased confidence-weighted self-deception (Cooperation vs. competition groups; $F(1, 620) = 15.0085$, $p_{bonf} = 3.442e\text{-}4$, $\eta_p^2 = 0.02673$)–people were more likely to confidently self-deceive to conform to their partners' bet in the cooperation group relative to defecting from the bet in the competition group. The absence of group by paranoia interaction emphasizes that in vs out-group membership was not differentially impacted by paranoia (Fig 3D and 3E). This contradicts the coalitional model of paranoia, which would predict increased self-deception in the high paranoia participants in the competition compared to the high paranoia participants in the cooperation group.

## Which trials engender self-deception?

Across paranoia groups, most self-deceptive responses occurred for the most ambiguous images (50/50 scene-face, Fig 3A). A GLME model showed a significant interaction between image ambiguity and paranoia group (GLME: $z = 5.853$, $p = 4.84e\text{-}09$, $b = 0.002643$, 95% CI: [0.0018, 0.00353])–the high paranoia group evinced self-deception to the slightly less ambiguous stimuli.

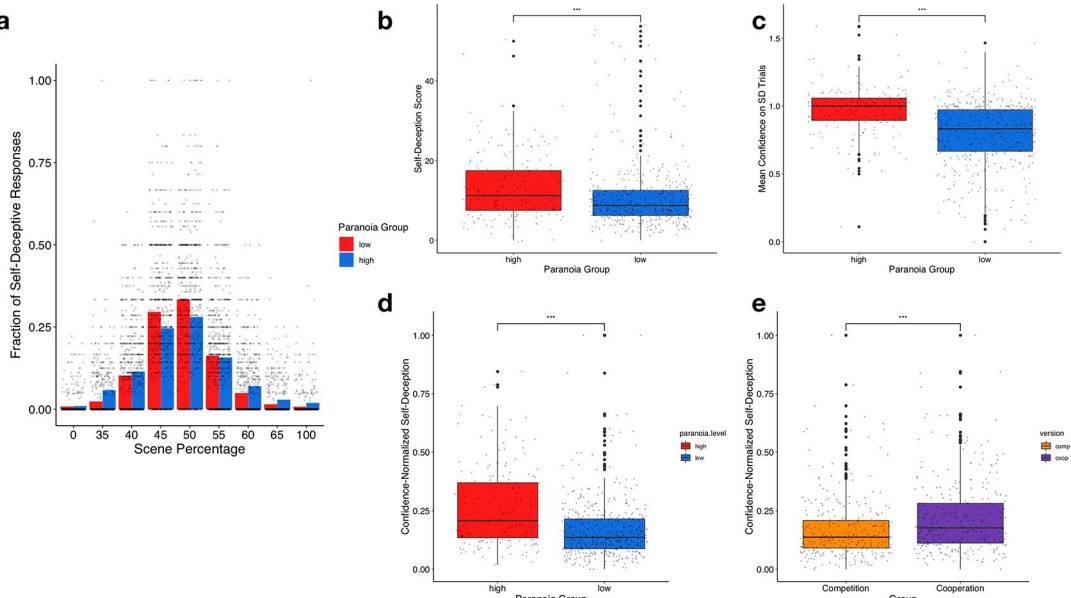

**Fig 3. Self-deceptive responses occurred more with ambiguous images and are different between paranoia groups. A**, the high paranoia group self-deceived more on slightly less ambiguous images than the low paranoia group. **B**, the high paranoia group had elevated raw self-deception scores (percentage of self-deceptive responses). **C**, mean confidence on those self-deceptive trials was elevated in high paranoia participants. **D**, the confidence-weighted self-deception, which controls for individual variation in baseline-confidence, is higher in the high paranoia group. **E**, confidence-weighted self-deception is also elevated in the cooperation group relative to the competition group. $^{*}P \le 0.05$, $^{**}P \le 0.01$, $^{***}P \le 0.001$.

## Computational modeling

The task is structured so that participants should ideally evaluate each image independently and ignore the bet. However, the existence of a motivational bias implies that the individual must be attributing some sort of value to the bet, and as a result, they might update this value as they gain more information through these trials. Participants might erroneously infer that image classification might depend on previous images, and update their beliefs about the images accordingly. In particular, these inaccurate associations might contribute to self-deceptive behavior.

To explore this idea, we decided to utilize a belief-updating model to draw inferences from participant choices regarding their latent beliefs about the task and stimuli. The Hierarchical Gaussian Filter (HGF) allows for the investigation of how hierarchical beliefs (in the perceptual model) influence choices (response model). The 2-arm HGF with integration of social and non-social information [28] allows us to separate the influence of the bet and the image and examine the higher level beliefs and parameters governing the perception of those cues. The generative model is outlined in Fig 2, and parameter descriptions can be found in S3 Table.

We found a significant difference in the initial beliefs (priors) at $x_{2,s}$ between the cooperation and competition groups. The cooperation group had a significantly elevated $\mu_{2,s}^0$ compared to the competition group (F(1, 654) = 16.7405, $p_{bonf}$ = 0.000145, $\eta_p^2$ = 0.03919; Fig 4A). The

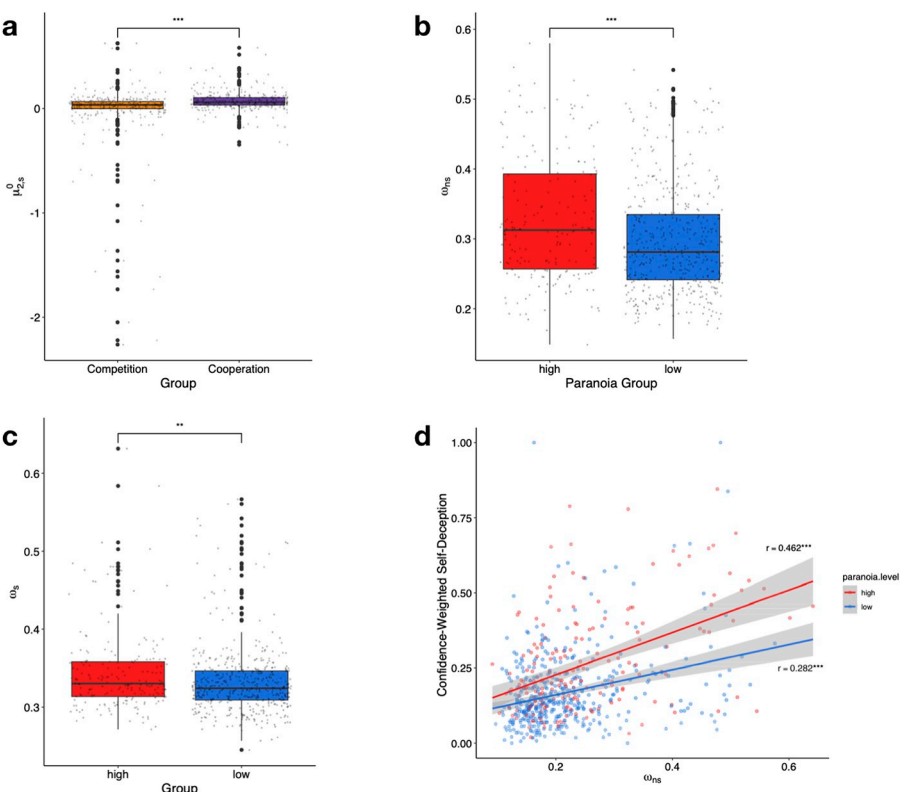

**Fig 4. Estimated parameters show differences based on paranoia group and experimental group. A**, the cooperation group has an elevated prior for the social information ($\mu_{2,s}^0$) compared to the competition group. **B**, the variance of the perceived tendency of image categorization ($\omega_{ns}$) is increased in high paranoia group as well as **C**, the variance of the perceived tendency of bet accuracy ($\omega_s$). *P ≤ 0.05, **P ≤ 0.01, ***P ≤ 0.001. **D**, $\omega_{ns}$ (variance of $x_{s,ns}$) is correlated with confidence weighted self-deception. The correlation is statistically stronger in the high paranoia group. *P ≤ 0.05, **P ≤ 0.01, ***P ≤ 0.001.

elevation in $\mu_{2,s}^0$ in the cooperation group represents a stronger initial belief that the bet would be more accurate, aligning with, and perhaps underwriting the observed motivational bias effect. This is also an important manipulation check, participants (regardless of their paranoia status) weighted the suggestion of a collaborator more strongly than that of a competitor. There was a significantly increased $\omega_{ns}$ in high paranoia participants ($F(1, 654) = 18.6837$, $p_{bonf} = 5.349\text{e-}5$, $\eta_p^2 = 0.027$) (Fig 4B). While $\omega_{ns}$ controls the variance of the second level belief, the interesting effect of this parameter is its interplay with the recency bias; a bias based upon the ambiguity of the previous image (Fig 2C). When $\omega_{ns}$ is greater, the recency bias term on the 1st level (the influence of the sensory inputs) impacts image classification less. This means that the ambiguity of the previous image has less impact upon the prediction for the next image, while the higher-level associations about the tendency of the image dominates the prediction. The sensory information contributes less to the classification while the higher-level associations contribute more–which is less optimal in a task where stimuli are independent. This could represent a lack of trust in one's abilities or sensory experiences which result in reliance upon the higher-level associative beliefs, independent of others' advice. The high paranoia group also evinced an elevated $\omega_s$ ($F(1,654) = 9.425$, $p_{bonf} = 0.006687$, $\eta_p^2 = 0.0133$)–so they showed increased variance of the second level belief governing the tendency of the bet to be accurate. Overall, this represents a more unstable belief about the perceived bet accuracy (Fig 4C). In both groups we found significant correlations between $\omega_{ns}$ and confidence-weighted self-deception (Fig 4D). While the low paranoia group evinced a significant correlation (Pearson's $r = 0.282$, $p = 1.535$–9), the correlation as significantly stronger in the high paranoia group (Pearson's $r = 0.462$, $p = 6.163\text{e-}11$, Fisher's z-transformed r, $p = 0.0228$). Self-deception independent of paranoia level was driven by $\omega_{ns}$ (perceived unreliability of ones' own choices). That drive was stronger in high paranoia participants.

## Bet manipulation

In order to better characterize the impact of social influences on perceptual decisions, we manipulated the accuracy of the bets in a follow up study (N = 324). In experiment 1, bets were 50% accurate. We increased bet accuracy to 75%. This manipulation significantly impacted self-deception and confidence. The number of self-deceptive trials and normalized confidence in self-deception decreased in the high paranoia group relative to experiment 1 (Independent samples t-test; raw self-deception: $t(192.12) = 3.0756$, $p_{bonf} = 0.004814$, Cohen's $d = 0.358$, 95% CI: [0.9612, 4.3981]; confidence: $t(96.42) = 2.5655$, $p_{bonf} = 0.02368$, Cohen's $d = 0.4157$, 95% CI: [0.01886, 0.1478]), while remaining unchanged in low paranoia participants (Independent samples t-test; raw self-deception: $t(679.42) = 2.1347$, $p_{bonf} = 0.06628$, Cohen's $d = 0.1488$, 95% CI: [0.0868, 2.076]; confidence: $t(461.33) = -0.934$, $p_{bonf} = 0.7016$, Cohen's $d = 0.0758$, 95% CI: [-0.05496116, 0.01954774]). This indicates that the high paranoia participants were sensitive to their partners' abilities (S3 Fig).

We fit the same 2-layer HGF to the new dataset. The difference in $\omega_{ns}$ in high paranoia we found in the original experiment was preserved in the follow-up ($F(1, 319) = 6.4532$, $p_{bonf} = 0.014$, $\eta_p^2 = 0.0208$), and there was no significant interaction between paranoia group and bet accuracy ($F(1, 977) = 0.451$, $p_{bonf} = 1$, S4C Fig). High paranoia participants evinced elevated variability in the tendency to perceive the image as face or scene, manifest as an overweighting of stimulus tendency rather than current sensory evidence. In contrast, there was no difference in $\omega_s$ ($F(1,319) = 0.0225$, $p_{bonf} = 1$, $\eta_p^2 = 0.000197$; S4B Fig) between paranoia groups. This suggests that increasing the partners' accuracy caused high paranoia participants to perceive less social volatility and behave less self-deceptively.

We found no difference between the two experiments in the number of trials on which participants could have self-deceived (instances in which bet differed from C1 classifications, Independent samples t-test: $t(532.55) = 0.21728$, $p = 0.8281$, Cohen's $d = 0.0159$, 95% CI: [-0.00617, 0.0077]). Furthermore, as in experiment 1, stimulus ambiguity drove self-deception–the 50/50 scene/face stimuli were most likely to engender self-deception. However, in experiment 2, the self-deception to less ambiguous cues was less pronounced. We replicated the effect of group manipulation (cooperation versus competition) on initial social beliefs. We found a main effect of experimental group, however, it did not survive Bonferroni correction for multiple comparisons ($F(1, 319) = 4.6709$, $p_{uncorrected} = 0.03112$, $p_{bonf} = 0.09336$, $\eta_p^2 = 0.014$; S4A Fig). Again, these prior beliefs were no different between the high and low paranoia participants. This lack of interaction is hard to reconcile with models of paranoia that rely on coalitional cognition, since we found no effect of paranoia on coalition or competition [7].

## Model selection and validation

For the model space shown in S4 Table, we compared a variety of perceptual and response models. Due to the high number of models, we used family-wise comparison to narrow down a winning perceptual and winning response model. Family BMS for the perceptual model space yielded a winning model of P1 (HGF with a scaled-$\omega_s$), with a protected exceedance probability of 1 (S5 Table). We found a winning response model of R1 (softmax with decision-noise only) with a protected exceedance probability of 0.9786 (S6 Table). Correspondingly, our winning model was M1, which used a P1 perceptual model and R1 response model.

## Model simulations

We utilized each individual parameter set found in Experiment 1 to simulate responses for each participant–the simulated responses were used to invert the original model to validate our findings regarding group differences in parameters. Each parameter of interest was significantly correlated with its simulated companion (S1 Fig: $\omega_{ns}$: $r = 0.2386473$, $p = 4.864e-10$; $\omega_s$: $r = 0.7283063$, $p < 2.2e-16$; log-transformed $\mu_{2,s}^0$; $r = 0.7815166$, $p < 2.2e-16$). The group differences based on paranoia group membership for both $\omega_{ns}$ ($F(1, 656) = 6.072$, $p = 0.014$, $\eta_p^2 = 0.0087$) and $\omega_s$ ($F(1, 656) = 15.95$, $p = 7.24e-5$, $\eta_p^2 = 0.0222$) were preserved in the simulated parameter sets, as well as the main effect of experimental group on the social priors ($\mu_{2,s}^0$; $F(1, 651) = 12.51$, $p = 0.000432$, $\eta_p^2 = 0.0186$). Successful parameter recovery and recapitulation of the observed group effects reassures us that we have the appropriate model.

## Bayesian versus non-Bayesian models

The Rescorla and Wagner (1972) rule centers prediction error in learning [30]. Cues have associations with valued outcomes and those associations are updated by mismatches between the associative predictions and the experienced outcomes (prediction errors) weighted by fixed associability parameters that correspond to the salience of the cues and outcomes [31]. Despite its success, the model is non-normative and heuristic [30]. It does not conform to the principles of probability theory and often performs poorly in real-world situations where outcomes and states must be inferred under uncertainty [30]. The Rescorla-Wagner rule had poor fit to our data, compared to the HGF-type models, even when accounting for model complexity (S4 and S5 Tables). A Bayesian model that incorporates uncertainty provided a better account of self-deception and overconfidence and their association with paranoia.

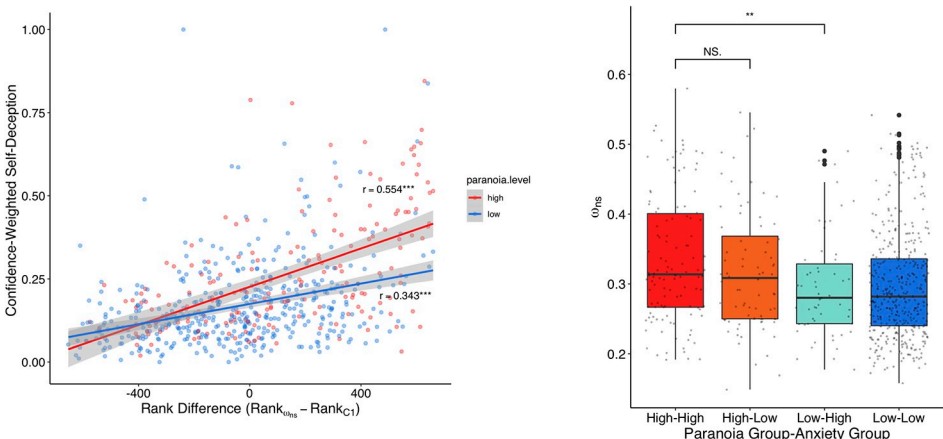

**Fig 5. A**, Difference between the relative perceived choice reliability (rank of $\omega_{ns}$) and relative objective classification performance (rank of C1 accuracy) is correlated with confidence-weighted self-deception. A higher $\omega_{ns}$ represents a lower perceived choice reliability (analogously, a higher perceived choice unreliability), so individuals scoring high on this rank difference have high perceived unreliability and low ability. This relationship is significantly stronger in the high paranoia group compared to the low paranoia group. **B**, High paranoia is responsible for elevated $\omega_{ns}$ independent of anxiety. The high paranoia high anxiety group showed similar values of $\omega_{ns}$ to the high paranoia low anxiety group, while the low paranoia high anxiety group had a significantly decreased $\omega_{ns}$.

## Self-esteem, paranoia & overconfidence

The initial classification phase (C1) measures each participant's objective classification ability. We ranked participants on this metric. Next, we ranked participants on their perceived choice reliability during C2 ($\omega_{ns}$). Computing the difference in these ranks gives a metric of participants' insight into their performance. Having a large difference in these ranks (rank of $\omega_{ns} \gg$ rank of C1-score) corresponds to an overly pessimistic view of oneself. We find a significant correlation between the rank difference and confidence-weighted self-deception (High paranoia: Pearson's $r = 0.554$, p = 6.163e-16; Low paranoia: Pearson's $r = 0.343$, p = 1.618e-13) which suggests that low self-confidence and diminished ability increase the incidence and confidence of self-deceptive responses (Fig 5A). These correlations were significantly different (Fisher's z-transformed r, p = 0.0027), suggesting paranoid participants take the opportunity to bolster their view of themselves. Overconfidence and self-deception protect against negative self-image, but at an economic cost.

## Demographics & confounds

Paranoia often correlates with demographic features and other affective states. We found significant correlations between depression and paranoia (Pearson's $r = 0.529$, p = <2e2-16), and between anxiety and paranoia (Pearson's $r = 0.612$, p = <2e2-16). In order to dissect the impact of anxiety and depression on the key model parameter ($\omega_{ns}$), we performed a multiple regression with self-ratings of paranoia, anxiety and depression as predictors. Both GPTS (paranoia) and BAI (anxiety) scores were significant predictors of $\omega_{ns}$ ($\beta_{GPTS} = 0.19$, 95% CI: [0.1, 0.29], t = 3.991, p = 7.34e-5; $\beta_{BAI} = 0.2$, 95% CI: [0.06, 0.34], t = 2.792, p = 0.00539), while BDI (depression) score was not ($\beta_{BAI} = -0.12$, 95% CI: [-0.25, 0.01], t = -1.766, p = 0.078). A Farrar-Glauber test for multicollinearity showed that there appeared to be collinearity between the BDI and BAI scores in particular (Overall collinearity: $\chi^2 = 981.0465$, p <0.05; Farrar-Glauber F-test: $F_{GPTS} = 381.9105$ (p <0.01), $F_{BDI} = 1239.6627$ (p <0.01), $F_{GPTS} = 1520.1804$ (p <0.01); Partial correlations: $\rho_{BDI,BAI} = 0.729$ (p <2.2e-16), $\rho_{GPTS,BAI} = 0.367$ (p <2.2e-16),

$\rho_{BDI,GPTS} = 0.07$ (p = 0.08). In order to dissociate the effects of anxiety and paranoia on these parameters, we split participants into four groups: Those with high paranoia and high anxiety (1), high paranoia and low anxiety (2), low paranoia and high anxiety (3), and low paranoia and low anxiety (4). We found that the high paranoia/high anxiety group (1) were no different from the high paranoia/low anxiety group (2) in $\omega_{ns}$ (F(3, 659) = 6.457, p = 0.00026,, $\eta_p^2$ = 0.0316, Post-hoc Tukey test, p = 0.345, 95% CI: [-0.0481, 0.0137]) while the low paranoia/high anxiety group (3) had a significantly lower $\omega_{ns}$ compared to the high paranoia/high anxiety group (1) (Post-hoc Tukey test, p = 0.0034, 95% CI: [0.00409, 0.0727]; Fig 5B). Though anxiety and paranoia are highly correlated, paranoia appears more responsible for the group differences in self-deception and the associated model parameters.

We performed ANCOVAs using demographics (race, ethnicity, age, gender), psychiatric diagnosis and medication usage, and socioeconomic factors (income, education) as covariates (S2 Table). All effects of paranoia group on $\omega_{ns}$ were robust to the inclusion of all the covariates, as was the effect of experimental group on initial beliefs.

## Discussion

People with high paranoia made more high-confidence self-deceptive responses during challenging perceptual decisions under social influence. They overrode their previous choices to agree with collaborators and defect from competitors. This effect was attenuated by making the partner's bets more accurate. We fit a computational model which captured how participants estimated and weighted the influence of current and historical sensory data as well as current and historical social inputs. In this framework self-deception in paranoia was not driven by changes in initial prior weighting of social information (though such priors did distinguish the group working with a collaborator from the group working against a competitor). Rather, the increased self-deception in high paranoia participants was driven by two processes: (1) an underweighting of current sensory inputs relative to the prevailing tendencies from recent trials and (2) an overweighting of the partners' current bet relative to the history of bet accuracy. Taken together, these data are consistent with self-deception flourishing in high paranoia as a result of a lack of confidence in ones' own perceptual inferences, coupled with an excessive influence of social suggestions (regardless of affiliation). We observed less self-deception when the partners' bets were more accurate, suggesting that self-deception is particularly likely in paranoid participants when self (non-social) and others (social) are experienced as unreliable sources of information.

Some have argued that motivated reasoning and self-deception contradict Bayesian accounts of belief updating, suggesting instead that biased beliefs are really preferences—things that people desire to be true, and that they are driven by identity (what defines people and their important groups like political parties) [32]. Others, have pushed back, suggesting instead that these biases might be understood in terms of differences in perceived reliability of evidence or evidence sources [33], prior beliefs [33], or deriving utility from beliefs and their consistency [34]. The HGF approach is inherently Bayesian [30,35], since it rests on sequential updating of beliefs according to Bayes' theorem, where beliefs represent inferences about hidden states of the environment (self, others, and external stimuli) in the form of posterior probability distributions, incorporating estimates of estimation uncertainty and environmental uncertainty [30,35]. Taking this approach, we found that over-confident self-deception and paranoia appears explainable in Bayesian terms: as changes in learning rates and relative weightings of social information, in response to pessimistic estimates about ones' own proficiency in perceptual judgments, particularly under high stimulus ambiguity. This model outperformed a simpler non-normative heuristic model [31] which neither fit nor simulated our

observations. Group identity drove changes in prior weightings, however, contrary to coalitional accounts of paranoia, we did not see those prior beliefs contributing significantly to self-deception and paranoia in our data. Neural data could further illuminate the issue of social and non-social contributions to belief updating and paranoia. For example, orbitofrontal cortex and amygdala may track non-social belief updating and dorsomedial and ventromedial prefrontal cortex more social specific mechanisms [36]. Our work suggests that paranoia may be the purview of the former, rather than the latter, though of course these mechanisms are densely interrelated [37–39].

In experiment 2, we found that decreasing the ambiguity of the social information (increasing the fidelity of the partner bets) was also impactful. Under social comparison theory, individuals are compelled to improve their performance and minimize discrepancies between their own and others' performance, generating competitive behaviour [40]. As we describe presently, uncertainty can prompt social comparison [40,41]. However, comparison concerns decrease dramatically when uncertainty about one's ranking relative to others is removed [42]. We contend that increasing the accuracy of partners' bets in experiment 2, neutralized high-confidence self-deception because it made the discrepancy between participant and partner performance clearer and rendered self-deception less necessary, warranted, or appropriate.

Our work involves online self-report of psychiatric symptoms. It is possible our high scoring participants were simply responding inattentively, and thus, our paranoid participants were not really paranoid but rather disengaged [43]. In the work establishing this concern, inattentive responders yielded depression and anxiety scores near the clinical mean, while our participants scored lower. Furthermore, we think it unlikely that inattentive responding (on tasks or scales) could yield the specific set of findings we report presently, rather, we imagine more random distribution of ratings across scales and choices across trial-type, instead of maximal self-deception during the most ambiguous trials. It is also hard to imagine how inattention would yield increased confidence on self-deceptive trials.

Our modeling work was consistent with self-deception impacting self-esteem and thence over-confidence in high paranoia participants. However, our task did not have a conduit for that over-confidence–in terms of convincing others of one's insights or abilities [3]. A task with reciprocal exchange between participants would be enlightening. Differing self-deception when confidence is communicated between partners would be consistent with a role for self-deception in deceiving others as well as self [1]. In an advice-giving task, patients with schizophrenia were overconfident in their own advice, particularly those with delusions [44]. Our data suggest this effect might be driven by self-deception secondary to an experience of one's own perceptual unreliability. Furthermore, boosting self-esteem–by conditioning positive self-associations—appears to mollify paranoia [45], it ought to similarly diffuse self-deception.

Given the debate about self-deception and delusions [46], it will be important to establish whether the same effects are present in people with confirmed delusional beliefs. Recent work on advice giving by people with schizophrenia suggests that patients with delusions are over confident in their advice [44]. We suggest that our data are consistent with the possibility that delusion (albeit on the extreme end of a continuum of paranoia) might entail self-deception. At the same time–in light of our data—delusion and self-deception may not violate epistemic rationality [47] and might harbor adaptive function [48].

## Supporting information

**S1 Table. Demographics table for experiment 1.**
(PDF)

**S2 Table. ANCOVAs for model parameters.**
(TIFF)

**S3 Table. Perceptual Model Parameters.**
(TIFF)

**S4 Table. Model Space.**
(TIFF)

**S5 Table. Family-wise Bayesian model selection for perceptual models.**
(TIFF)

**S6 Table. Family-wise Bayesian model selection for response models.**
(TIFF)

**S7 Table. Initial Prior Values for parameters.**
(TIFF)

**S1 Fig. Parameters of interest were correlated with their simulated values obtained from simulation of responses and inversion of the model.**
(TIFF)

**S2 Fig.** Fraction of self-deceptive responses using the simulated responses for the winning model (**A**) and for a simulated responses of a normative model (**B**). The normative model simulations fail to capture the key components of the behavioral data–it predicted too many self-deceptive responses for the extreme images and did not properly predict the differences based on paranoia group.
(TIFF)

**S3 Fig. Self-deception and confidence-weighted self-deception is different between the two experiments only in the high paranoia group. A**, raw self-deception scores are lower in the high paranoia group with more accurate bets than the high paranoia group with less accurate bets. There is no difference in the low paranoia group. **B**, mean confidence on self-deceptive trials decreases in the high paranoia group with higher bet accuracy. $^{*}P \leq 0.05$, $^{**}P \leq 0.01$, $^{***}P \leq 0.001$.
(TIFF)

**S4 Fig. Group parameter differences in the two experiments differ. A**, both experiments show a difference in social priors ($\mu^0_{2,s}$), with the cooperation group having an increased prior compared to the competition group. **B**, the difference in the variance of the 2nd level belief on the social side ($\omega_s$) between paranoia groups disappears when the bet accuracy is higher in experiment 2. **C**, the difference between paranoia groups in the variance of the 2nd level belief about the image ($\omega_{ns}$) is maintained in both experiments. $^{*}P \leq 0.05$, $^{**}P \leq 0.01$, $^{***}P \leq 0.001$.
(TIFF)

## Author Contributions

**Conceptualization:** Rosa A. Rossi-Goldthorpe, Philip R. Corlett.

**Data curation:** Rosa A. Rossi-Goldthorpe.

**Formal analysis:** Rosa A. Rossi-Goldthorpe, Pantelis Leptourgos.

**Funding acquisition:** Philip R. Corlett.

**Investigation:** Rosa A. Rossi-Goldthorpe, Pantelis Leptourgos, Philip R. Corlett.

**Project administration:** Rosa A. Rossi-Goldthorpe.

**Resources:** Yuan Chang Leong, Philip R. Corlett.

**Software:** Yuan Chang Leong.

**Supervision:** Pantelis Leptourgos, Philip R. Corlett.

**Writing – original draft:** Rosa A. Rossi-Goldthorpe, Yuan Chang Leong, Pantelis Leptourgos, Philip R. Corlett.

**Writing – review & editing:** Rosa A. Rossi-Goldthorpe, Yuan Chang Leong, Pantelis Leptourgos, Philip R. Corlett.

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
