## [Decision Letter · Decision Letter 0]

21 Jul 2021

Dear Dr. Corlett,

Thank you very much for submitting your manuscript "Paranoia, Self-Deception, and Overconfidence" for consideration at PLOS Computational Biology.

As with all papers reviewed by the journal, your manuscript was reviewed by members of the editorial board and by several independent reviewers. In light of the reviews (below this email), we would like to invite the resubmission of a significantly-revised version that takes into account the reviewers' comments.

The three reviewers all saw quite some merit in your paper. However, two of them had serious concerns, that mostly involved a lack of clarity in the model, the behavioural variables measured and the statistical tests done. Reviewer 2 also makes the important point that the hypotheses should be clear from the introduction, which I second. Finally, reviewer 3 brings up an important potential confound of inattention. I therefore would like to request you to carefully go through each of the points made by the reviewers and to revise the manuscript to clarify all these points.

We cannot make any decision about publication until we have seen the revised manuscript and your response to the reviewers' comments. Your revised manuscript is also likely to be sent to reviewers for further evaluation.

Sincerely,

Marieke Karlijn van Vugt, PhD

Associate Editor

PLOS Computational Biology

Samuel Gershman

Deputy Editor

PLOS Computational Biology

Reviewer's Responses to Questions

**Comments to the Authors:**

Reviewer #1: This is an excellent and timely paper. The authors set out to investigate the interaction of social context (cooperation vs competition) with perceptual belief updating and trait paranoia. To do so they designed an elegant joint perceptione experiment in which participants view visual morphs and make bets as to which morph is dominate, either competing through deception or cooperating through self-deception. The authors then apply this task in a large online cohort, and model responses using an adapted version of the hierarchical gaussian filter, operationalizing the decision process as a dual stream processes in which social and perceptual beliefs are integrated together. This approach reveals interesting differences in choice behavior and underlying computational mechanisms, which are further replicated and extended in a second follow up experiment manipulating choice accuracy. Overall the findings are robust and interesting, the modelling appropriate, and the manuscript extremely well written. I have little to add to this manuscript other than a curiosity about the confidence ratings, which here are largely used to titrate behavioral responses. However, adding metacognitive modelling at this stage would probably bloat an otherwise extremely tight paper, so this is just my curiosity. I have only one major comment for consideration:

1. If I understand the model validation steps correctly, the authors do an excellent job of motivating a complex family of models determining the complexity of the winning dual stream model. They also compare this model to a non-bayesian alternative using model comparison, and then conduct a posterior predictive check by correlating simulated data generated by this model to empirically fitted data. My only question is if this final step is sufficient - from the supplementary figure there appears to be some potential deviance between the two parameter steps. Is there a way to quantify this directly? I could imagine for example using a cross-validation procedure, e.g. fitting the models to a subset of the data and then predicting "hold out" data to determine empirical out of sample prediction accuracy. However, this may be unnescessary if I have misunderstood some aspect of the model validation procedure.

Congratulations to the authors for an excellent contribution - good work.

Reviewer #2: Rossi-Goldthorpe RA et al.

The authors describe data from a study in which they examined perceptual inference in participants under conditions in which they did or did not get advice from a confederate who was either collaborating or competing with them. Data was analyzed at both the direct behavioral level and with an HGF.

I think this study has a lot going for it. The basic task design and the results (as far as I can sort them out) are quite interesting. In general this is also an interesting question of relevance to psychopathology. However, many things are not clearly explained. I had a hard time understanding the task design, the behavioral variables being analyzed and, more fundamentally, the hypotheses and how the overall approach was addressing those. Although I could understand enough to generally follow the thread of the results, I think a much more detailed account of the task, the behavioral data analyzed, and a more clear account of the findings would substantially improve this manuscript.

Comments

1. The details of the task were unclear, as was the definition of self-deception. For some of the ANOVA models it wasn’t clear what the dependent variable was. For example, “Analysis of variance revealed a main effect of paranoia (high or low), a main effect of group (competition or collaboration) but no paranoia by group interaction for self-deception.” I don’t know what is being analyzed. I think the behavioral data should be shown more clearly. And then the metrics that are derived from that data that serve as dependent variables should be explained clearly and illustrated. Best to do this in the results, but the methods could also use more detail on these points.

2. It would be useful to show and reference a figure for this when it is stated, “the probability of responding scene followed an s-shaped psychometric curve, indicating that in general, participants were able to categorize the chimeras accurately.”

3. For this section, “The difference between groups remained significant when we examined confidence-normalized self-deception score (F(1, 620) = 58.0612, pbonf= 2.8659e-13,   2 184 = 0.0859; Figure 3B, C). We also found that the cooperation group had increased confidence-weighted self-deception (F(1, 620) = 15.0085, pbonf=3.442e-4,   2 186 = 0.02673) – people were more likely to confidently self-deceive to conform to their partners’ bet in the cooperation group relative to defecting from the bet in the competition group. The absence of

group by paranoia interaction, suggests that centering in vs out-group membership was not differentially impacted by paranoia (Figure 3D-E).” What is confidence-normalized self-deception score? What is “centering in vs out-group…”?

4. The model should also be explained in the results. Some comments on what the variables measure. I know some of this is in the methods, but it would be much easier to give some background on the model and the variables in the results. The technical definitions can be given in the methods. It’s best to present the results and then summarize what they mean in each paragraph. It is also helpful if you can go back to the raw data and compute metrics directly on the raw data that show the effects extracted by the model. Also, I assume the model is mainly for studying learning, but the learning effects were buried and so it was not clear how participants updated belief estimate over trials on the basis of past feedback. It is also not clear why there was a comparison to Rescorla-Wagner, since such a model has no way of incorporating ambiguity in the cues, and the RW model is also no really applicable since the cues contain the information about the relevant behavior, whereas RW is more learning associations between values and arbitrary cues. This is not really a meaningful comparison.

5. The introduction could use a rewrite. It does not flow well. But more importantly, it does not setup the specific experiment that was done, or why it was done. What are the hypotheses and how does the experiment address these?

Reviewer #3: This is an interesting study of how perceptual beliefs can be swayed by the view of a confederate, in subjects with low or high paranoia levels. The authors used an online task in which a chimeric image is shown and a participant must rate it as majority face or house. A confederate then 'bets' on the outcome in a second condition, and the rating is made again, with a bonus for both being correct in the collaborator condition, or for correctly identifying they are incorrect in the competitor condition. A Bayesian model was used to evaluate the relative contributions of previous trials and social information to decisions made.

The authors found:

- Participants were more likely to agree with collaborators' bets and disagree with opponents' bets

- This was especially the case for high paranoia participants (but not when bets were more accurate), but no interaction with competition/collaboration was found

- The modelling found

the cooperation group had stronger initial faith in the bet

lower recency bias (bigger omega_ns) about the image in high paranoia (replicated in another sample with higher bet accuracy)

>updating about the accuracy of the bet (bigger omega_s) in high paranoia

- Lower recency bias about the image correlated with 'self-deception' - this was stronger in high paranoia

- Lower recency bias appeared to relate to paranoia rather than anxiety, although these were correlated

Overall the paper uses a clever task and sophisticated modelling to show that paranoia is associated with less trial-to-trial influence of the image, and more influence of the confederate (in terms of updating about bet accuracy). I do have some concerns about the results, however: the main ones are i) whether this pattern of results could also be explained by inattentive responding? ii) I find some of the statements in the paper rather strong given the evidence - especially the term 'self-deception', and some assertions about paranoia. These are detailed below:

p3 - Were the subjects told that the reward-maximising strategy is to judge accurately?

Also, describing a response that changes according to the opponent/collaborator's bet as "self-deceptive" doesn't make sense to me. If I adjust my opinions having heard someone else's views, am I deceiving myself? This seems too strong and loaded a term to apply to this effect. For example, on p7, the finding that "self-deception... was driven by perceived unreliability of ones' own choices" could better be described as "loss of confidence in oneself increases the influence of others".

p4 - The description of the model is quite confusing and scanty. It needs to be much clearer. What is zeta? What is eta? What is w in Figure 2A? Is one of these the recency bias, as that term doesn't appear in Supp Table 3? Omega_ns doesn't appear in any equation that I found - yet it is the parameter behind the key group effect?

p7 - Given there was no trial-to-trial dependency in this task, it seems unwarranted to say that a lower recency bias in paranoid individuals might reflect a "lack of trust in one's abilities", given that having no recency bias at all is actually optimal here, if I understand correctly?

Also, a persuasive recent preprint by Zorowitz et al (psyarxiv.com/rynhk) showed that inattentive responding can induce spurious results in online studies. Can the authors reassure the reader that this could not explain their results? It seems to me that more inattentive players are likely to show less influence of recent trials, more influence of a confederate, and are more likely to score highly on anxiety/paranoia scales. The fact that no interaction with competition/collaboration is a bit concerning here - an interaction would make the results more specific to paranoia itself. Are any of these effects highly unlikely to be accounted for by inattentive responding?

p8 - What exactly would the 'coalitional cognition' account of paranoia predict, and why?

p9 - Why is it not the case that high paranoia individuals just have a more uncertain model? Hence self-confidence is lower and ability is diminished, but others' influence over them is stronger? What exactly justifies the added interpretation that overconfidence and 'self-deception' are *protecting* against negative self-image? Rather than just inevitable consequences of having a noisier model?

p10 - It is odd to perform a multiple regression showing that both paranoia and anxiety significantly correlate with omega_ns, but not to report the actual results in terms of betas, confidence intervals etc. If the regression is abandoned because of multicollinearity issues, the variance inflation factors should be reported to justify this. What was the cut-off for 'high' vs 'low' anxiety and how was it chosen? Also, the authors state "paranoia appears more responsible for the group differences in self-deception and the associated model parameters" - but omega_ns (unless I misunderstand) concerns the recency bias, not self-deception: its relationship with self-deception is indirect. To make the statement above, this analysis should have looked at self-deception, paranoia and anxiety directly?

p10 - the first process driving increased self-deception in high paranoia is described as "an underweighting of current sensory inputs relative to the prevailing tendencies from recent trials", but to me a reduced recency bias in that group ought to mean current inputs are *less* influenced by prevailing tendencies?

p11 - is it not too much to assert that "we found... paranoia can indeed by explained in Bayesian terms" given that no interaction with condition was found? i.e. the effect of competition was not greater in these subjects? Surely an account of paranoia must explain the particular direction of the effect?

Minor points

p2 - I cannot find the Hagen (2008) reference but does it really show that too much self-deception leads to delusional beliefs? Also the evidence that paranoia protects self-esteem is pretty weak, all told (Murphy et al, 2018, Lancet Psych), so the claims about paranoia causing "direct inflation of self-image" are not realistic.

Supp Fig 1 - the correlations are not reported? Also the recovered parameters end up squashed into a smaller range, although I suppose this doesn't matter so much if one is only interested in group differences. Also for mu0_2, this should be transformed and the correlation computed in transformed space as the vast majority of the datapoints are in the cloud just above zero.

**Have the authors made all data and (if applicable) computational code underlying the findings in their manuscript fully available?**

Reviewer #1: Yes

Reviewer #2: Yes

Reviewer #3: Yes

PLOS authors have the option to publish the peer review history of their article (what does this mean?). If published, this will include your full peer review and any attached files.

Reviewer #1: **Yes: **Micah Allen

Reviewer #2: No

Reviewer #3: No
---

## [Decision Letter · Decision Letter 1]

8 Sep 2021

Dear Dr. Corlett,

Thank you very much for submitting your manuscript "Paranoia, Self-Deception, and Overconfidence" for consideration at PLOS Computational Biology. As with all papers reviewed by the journal, your manuscript was reviewed by members of the editorial board and by several independent reviewers. The reviewers appreciated the attention to an important topic. Based on the reviews, we are likely to accept this manuscript for publication, providing that you modify the manuscript according to the review recommendations.

The two reviewers who were not yet satisfied with your manuscript are now a lot happier, but still request a few minor revisions. I suggest you carefully look at these suggestions to clarify the writing even further. Good luck!

Sincerely,

Marieke Karlijn van Vugt, PhD

Associate Editor

PLOS Computational Biology

Samuel Gershman

Deputy Editor

PLOS Computational Biology

[LINK]

Reviewer's Responses to Questions

**Comments to the Authors:**

Reviewer #2: The authors have substantially improved the manuscript. The intro could still be written a bit more fluently, particularly the first paragraph. But it’s better. I am still not clear on a few details of the experiment.

Specifically, I assume that in C1, there is no partner response? The participant is making choices but they do not see a choice made by a partner? This should be made clear. I assume there was no partner, but then I did not understand this sentence, “In experiment 1, the bets were correctly exactly 50% of the time.” Is this referring to the partner’s bets in C2? Or what is this referring to?

The payoff matrix should be stated in the methods, with a reference to Figure 1B. This is important. Also, in the 50% condition, it is not clear how the subject should respond. Specifically, I don’t think the following comment is clear for the 50% condition, “Crucially, the reward maximizing strategy is to classify the images correctly.” This would only be true if the partner really was guessing and therefore correct at chance level. So, the subject has to assume this. If the subject assumes the partner is above chance it would make sense to respond with or against the partner depending on the condition. It is possible then that paranoia changes the subject’s assessment of whether the partner is correct or not, and the subject is behaving optimally, as opposed to in a self-deceptive way.

It is possible that the experimental design accounts for this possibility, but if it does, it was not clear to me. Please clarify why the subjects should not respond against the partner in the competition condition and with the partner in the cooperation condition, when the stimulus is ambiguous (50%). Please also clarify why paranoia could not also be affecting the subject’s assessment of the partner’s accuracy.

Reviewer #3: Thanks to the authors for responding to my comments - I have only some minor remaining points that don't require re-review.

Regarding the concept of 'self-deception' in this experiment. I had not grasped that the partner is betting on an image *that they have not seen either* - can this be made as clear as possible in the methods? I had assumed the partner had seen it but the player had not.

The addition to the discussion: "Paranoia was found to be unrelated to betrayal aversion – when one has a higher aversion to risky situations where outcomes are contingent upon social factors compared to non-social factors which does support the coalitional cognition model". Is this correct or do the authors mean this does NOT support the coalitional model? I am not sure why absence of betrayal aversion supports that model?

About the parameter recovery - I hope it does not become common practice to only report p-values: these will be strongly affected by the number of simulations conducted and so are a bit meaningless. The correlation is really the key measure.

**Have the authors made all data and (if applicable) computational code underlying the findings in their manuscript fully available?**

Reviewer #2: Yes

Reviewer #3: Yes

PLOS authors have the option to publish the peer review history of their article (what does this mean?). If published, this will include your full peer review and any attached files.

Reviewer #2: No

Reviewer #3: No

Figure Files:

Data Requirements:

Reproducibility:

References:

---

## [Editor Report · Decision Letter 2]

15 Sep 2021

Dear Dr. Corlett,

We are pleased to inform you that your manuscript 'Paranoia, Self-Deception, and Overconfidence' has been provisionally accepted for publication in PLOS Computational Biology.

Best regards,

Marieke Karlijn van Vugt, PhD

Associate Editor

PLOS Computational Biology

Samuel Gershman

Deputy Editor

PLOS Computational Biology

I think you have sufficiently addressed the reviewers' comments. Congratulations with the acceptance of your paper.

---

## [Editor Report · Acceptance letter]

30 Sep 2021

PCOMPBIOL-D-21-00920R2 

Paranoia, Self-Deception, and Overconfidence

Dear Dr Corlett,

I am pleased to inform you that your manuscript has been formally accepted for publication in PLOS Computational Biology. Your manuscript is now with our production department and you will be notified of the publication date in due course.

With kind regards,

Andrea Szabo
